# De novo atomic protein structure modeling for cryoEM density maps using 3D transformer and HMM

Nabin Giri [1,2] & Jianlin Cheng [1,2] ✉

Accurately building 3D atomic structures from cryo-EM density maps is a crucial step in cryo-EM-based protein structure determination. Converting density maps into 3D atomic structures for proteins lacking accurate homologous or predicted structures as templates remains a significant challenge. Here, we introduce Cryo2Struct, a fully automated de novo cryo-EM structure modeling method. Cryo2Struct utilizes a 3D transformer to identify atoms and amino acid types in cryo-EM density maps, followed by an innovative Hidden Markov Model (HMM) to connect predicted atoms and build protein backbone structures. Cryo2Struct produces substantially more accurate and complete protein structural models than the widely used ab initio method Phenix. Additionally, its performance in building atomic structural models is robust against changes in the resolution of density maps and the size of protein structures.

Determining the three-dimensional (3D) atomic structures of macromolecules, such as protein complexes and assemblies[1–3], is fundamental in structural biology. The 3D arrangement of atoms provides essential insights into the mechanistic understanding of molecular function of proteins[4]. In recent years, cryo-electron microscopy (cryo-EM)[5] has emerged as a key technology for experimentally determining the structures of large protein complexes and assemblies[6–8]. However, modeling atomic protein structures from high-resolution cryo-EM density maps, which constitute a significant portion of the maps deposited in the EMDB[9], is both time-consuming and challenging, especially in the de novo setting when accurate homologous or predicted structures for target proteins or their units (chains) are unavailable[10,11]. Modeling atomic protein structures from cryo-EM maps faces the challenges of identifying atoms of proteins in density maps as well as tracing the atoms into chains to form the backbone structures and registering amino acid sequences with them[12].

Despite the importance of the problem, only a small number of methods have been developed for determining atomic structures from cryo-EM maps, such as Phenix[13], DeepMainmast[12], DeepTracer[14], and ModelAngelo[15]. Phenix is the most widely used standard tool of building atomic protein structures from cryo-EM density maps using classic molecular optimization. DeepTracer provides a web-based deep learning tool for users to predict atomic structures from density

maps. ModelAngelo combines information from cryo-EM map data, amino acid sequences, and prior knowledge about protein geometries to refine the geometry of the protein chain and assign amino acid types. DeepMainmast, a recently developed method, integrates AlphaFold2[16] with a density tracing protocol to determine atomic models from cryo-EM maps. Incorporating accurate AlphaFold-predicted structures into the modeling has significantly improved the quality of the structures determined from cryo-EM density maps[12].

However, modeling multi-chain protein structures from cryo-EM density maps remains a challenging task for the existing methods, particularly when there are inaccurately predicted structures for target protein complexes or their chains to be used as templates. The de novo modeling of protein structures from only density maps without using templates is not only practically relevant in this situation, but also can help answer an important question: how much structural information can be extracted from cryo-EM density maps alone? In the de novo modeling context, we introduce Cryo2Struct (i.e., cryo-EM to structure), a fully automated, ab initio modeling method that does not require predicted or homologous structures as input to generate 3D atomic structures from cryo-EM density maps alone. Cryo2Struct first uses a Transformer-based deep learning model with an attention mechanism[17] to identify atoms and their amino acid types in cryo-EM density maps. Then it uses an innovative generative Hidden Markov

[1]Department of Electrical Engineering and Computer Science, University of Missouri, Columbia, MO, USA. [2]Roy Blunt NextGen Precision Health, University of Missouri, Columbia, MO, USA. ✉e-mail: chengji@missouri.edu

Model (HMM)[18] and a tailored Viterbi Algorithm[19] to align protein sequences with the predicted atoms and amino acid types to generate atomic backbone structures. Cryo2Struct is rigorously tested on 628 density maps in the stringent ab initio modeling setting in which no homologous/predicted structure is used as a template and yields substantially improved modeling accuracy.

## Results

### Atomic structure modeling workflow

Cryo2Struct takes a 3D cryo-EM density map and the corresponding amino acid sequence of a protein as input to generate a 3D atomic protein structure as output automatically (Fig. 1a–e). As in ref. 20, we divide the problem of atomic structure determination from cryo-EM density map into an atom classification (recognition) task and a sequence-atom alignment task. The two tasks are performed by a Deep Learning (DL) block based on a transformer (Fig. 1b) and an alignment block based on a HMM (Fig. 1d), respectively. The DL block classifies each voxel (3D pixel) within the cryo-EM density map into different types of backbone atoms (e.g., Cα) or non-backbone voxel and predicts their amino acid types, while the alignment block constructs a HMM[18] from predicted Cα atoms (corresponding to the hidden states in the HMM) and aligns the amino acid sequences with them using a customized Viterbi Algorithm, resulting in a sequence of Cα atoms connected as protein chains to form the atomic backbone structure of the protein. Additional details are available in the "Methods" section.

### Predicting backbone atom and amino acid types using 3D transformer

The first step of the atomic structure modeling is to detect the voxels in cryo-EM density map that contain backbone atoms and predict their amino-acid types. We designed and trained a 3D transformer-based model to classify each voxel of the cryo-EM density map into one of four different classes representing three backbone atoms (Cα, C, and N), and the absence of any backbone atom. Another 3D transformer-based model was designed and trained to classify each voxel of the cryo-EM density map into one of twenty-one different amino acid

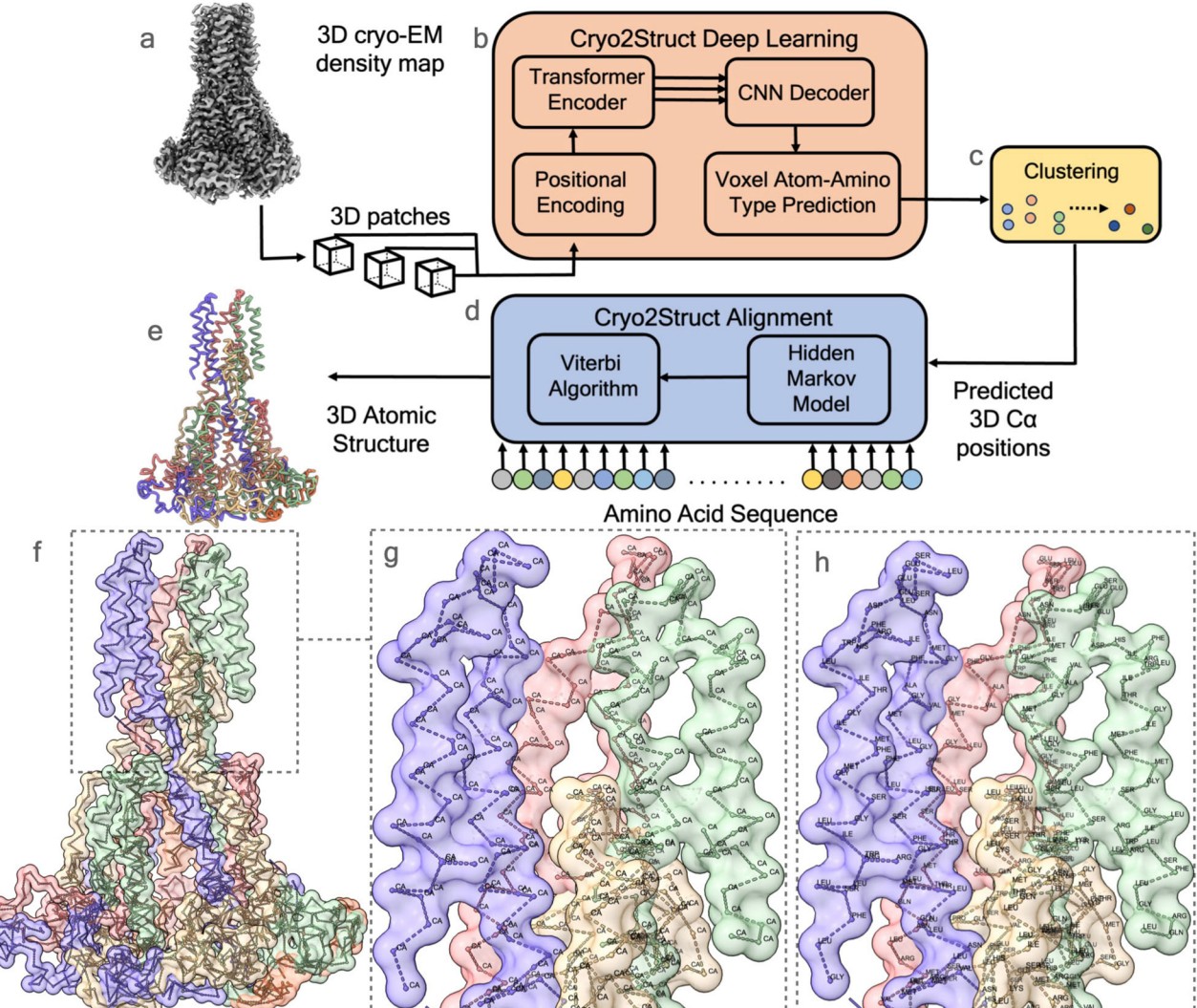

**Fig. 1 | An overview of the automated prediction workflow of Cryo2Struct.** Given a 3D cryo-EM density map of a protein as input (**a**), the Deep Learning block based on a transformer (**b**) generates a voxel-wise prediction of Cα atoms and their amino acid type. A clustering step (**c**) is used to merge nearby predicted Cα atoms into one atom to remove redundancy. The predicted Cα atoms and their amino acid type probabilities are used by the Alignment block (**d**) to build a Hidden Markov Model (HMM), which is used by a customized Viterbi Algorithm to align the sequence of the protein with it to generate a 3D backbone atomic structure for the protein (**e**). **f** shows the skeleton of the Cryo2Struct modeled structure for a test cryo-EM density map having less than 25% sequence identity with the training data released on 13 September 2023 (EMD ID: 41624; resolution 2.8 Å), where each chain is colored differently. **g** depicts the connected Cα atoms, and **h** shows the amino acid types assigned to the Cα atoms; the modeled structure has 1585 amino acid residues; and the F1 score of Cα atom prediction is 89.1%. Source data are provided as a Source Data file.

classes representing twenty standard amino acids and the absence of an amino acid or unknown amino acid. The models were trained as a sequence-to-sequence predictor, utilizing a Transformer-Encoder[17] to capture long-range voxel-voxel dependencies and a skip-connected decoder to combine the extracted features at different encoder layers to classify each voxel. The models were trained using the large Cryo2StructData dataset[21]. Cryo2StructData is a comprehensive labeled dataset of cryo-EM density maps curated specifically for deep learning-based atomic structure modeling in cryo-EM density maps. The models were trained and validated on the entire dataset comprising 6652 cryo-EM maps for training and 740 cryo-EM maps for validation first and then were blindly tested on two test datasets. Because some predicted C$\alpha$ voxels are spatially very close and likely correspond to the same C$\alpha$ atom, Cryo2Struct employs a clustering strategy to group predicted C$\alpha$ voxels within a 2Å radius into clusters and select the centrally located C$\alpha$ voxel in each cluster as the final predicted C$\alpha$ atom (see the Method Section for details).

### Aligning protein sequence with predicted C$\alpha$ atoms

The goal of this step is to connect the predicted, disjoint C$\alpha$ atoms into peptide chains and assigns amino acid types to them (sequence registering). To achieve the goal, the alignment block constructs a HMM from the predicted C$\alpha$ atoms and their predicted amino acid type probabilities, in which each predicted C$\alpha$ atom is represented by one hidden state. The transition probability between two hidden states is assigned according to the spatial distance between their corresponding C$\alpha$ atoms, and the emission probability of each hidden state for generating 20 different amino acids is assigned according to the predicted probability of 20 different amino acid types for its C$\alpha$ atom (see more details in Methods Section). The sequence of each chain of the protein is aligned to the HMM by a customized Viterbi algorithm to generate the most probable path of hidden states (C$\alpha$ atoms). The path for a chain represents the connected C$\alpha$ atoms of its backbone structure. The paths for multiple chains of a protein, together with the sequences aligned with them, form the final atomic backbone structure of the protein. Figure 1f illustrates a high-quality structure modeled by Cryo2Struct, while Fig. 1g, h provides a detailed view of the structure. In Fig. 1g, the predicted C$\alpha$ atoms are depicted and connected by the alignment block. Fig. 1h reveals the amino acid-type assignment for each C$\alpha$ atom.

### Comparing Cryo2Struct with Phenix on a standard dataset

After Cryo2Struct was trained and validated, we first compared the modeling performance of Cryo2Struct and Phenix[13] on a standard test dataset that was used to benchmark Phenix's map_to_model tool[22]. Most density maps in the dataset are for multi-chain protein complexes, while some of them are associated with single-chain proteins. Their resolution ranges from 2.08 Å to 5.6 Å. The average resolution of the density maps is 3.68 Å. The number of amino acid residues included in the maps varies from 448 to 8,416. These test maps were not present in the training and validation dataset used to train the Cryo2Struct DL model. We chose Phenix as a reference here because it built the structures from the density maps in the same ab initio mode as Cryo2Struct is designed to do without using homologus or predicted protein structures as input. The structures built by Phenix were downloaded from its website[22]. The structural models built for the 128 test cryo-EM maps in the test data by Cryo2Struct and Phenix were compared with the true structures in the Protein Data Bank (PDB) to evaluate their quality. The evaluation results in terms of six metrics are presented in Fig. 2.

Figure 2a plots the recall of C$\alpha$ atoms of each model built by Cryo2Struct for each of the 128 density maps against that by Phenix. The recall (sensitivity) represents the fraction of actual C$\alpha$ atoms in the true structure that are correctly identified by a model. Cryo2Struct achieves an average recall score of 65%, much higher than 40% of

Phenix, indicating that Cryo2Struct recovers a much higher percentage of C$\alpha$ atoms correctly than Phenix. On 126 out of 128 density maps, Cryo2Struct has a higher recall than Phenix.

Figure 2b plots the F1 score of C$\alpha$ atoms of Cryo2Struct against Phenix. The F1 score is the harmonic mean of precision and recall of C$\alpha$ atoms. The precision (specificity) is the percentage of predicted C$\alpha$ atoms that are correct ones. The F1 score is a balanced measure because it considers both the specificity and sensitivity of predicted C$\alpha$ atoms. The average F1 score of Cryo2Struct and Phenix is 66% and 52%, respectively. On 105 out of 128 maps, Cryo2Struct has a higher F1 score.

Figure 2c plots the global normalized TM-scores of the models built by the two methods. A standard TM-score measures the similarity between a model and the corresponding known structure, which was calculated by a protein complex structure comparison tool - US-align[23] by enabling its options for aligning two multi-chain oligomeric structures and all the chains, as recommended for aligning biological assemblies. In this analysis, to fairly compare the models built by Cryo2Struct and Phenix that usually have different lengths (numbers of residues), the global TM-score is normalized by the same length of the experimental structure. The TM-score ranges from 0 to 1, with 1 being the best possible score. The average global normalized TM-score of Cryo2Struct is 0.2, more than double 0.084 of Phenix. On 114 out of 128 density maps, Cryo2Struct has a higher normalized TM-score than Phenix.

However, the average global normalized TM-score of both methods is still low. One reason is that the TM-score is a sequence-dependent global measure and obtaining a high normalized TM-score requires a high portion of C$\alpha$ atoms of a large protein complex being not only correctly identified (high recall) but also all correctly linked at the same time, which is still very challenging for the de novo atomic model building from only the density maps that may have missing density values in some regions causing disconnection of C$\alpha$ atoms. Another reason is that the TM-score computed by US-align is normalized by the total length of the known structure, which is usually very large (average length of the true structure = 3794.95 residues) rather than the length of a structurally aligned region between a model and the true complex structure. So, if the aligned region has a high TM-score but is only a fraction of the entire known structure, the normalized TM-score would still be low. We expect that complementing density maps with the features extracted from protein sequences or AlphaFold-predicted structures as input for deep learning to predict C$\alpha$ atoms and amino acid types can further improve the normalized TM-score[10,12].

Figure 2d compares the aligned C$\alpha$ length of the structural models built by Cryo2Struct and Phenix, which was computed by US-align. The aligned length is the number of C$\alpha$ atoms denoting residues that have been successfully matched or aligned between the predicted model and its true structure. The average length of the true structures for all 128 test maps is 3794.95. The average aligned length of Cryo2Struct's models is 945.55 (about 24.9% of the length of the known structure on average), 2.6 times the average length 358.51 of Phenix (about 9.4% of the length of the known structure on average). On 120 out of 128 density maps, CryoStruct has a larger aligned length than Phenix. Another interesting phenomena is that the models constructed by Cryo2Struct always have the same or very similar number of residues as the corresponding true structures (supplementary Fig. S1a) and therefore capture the overall shape of the true protein structure well despite of some errors in the local regions and atom connections, while the models constructed by Phenix usually are much smaller than the true structures (supplementary Fig. S1b) and therefore only renders a portion of the true structures.

In addition to using US-align to compare the models with the known structures, we also used the phenix.chain_comparison tool to compare a model and the true structure to compute the percentage of matching C$\alpha$ atoms, as shown in Fig. 2e. It calculates the C$\alpha$ match

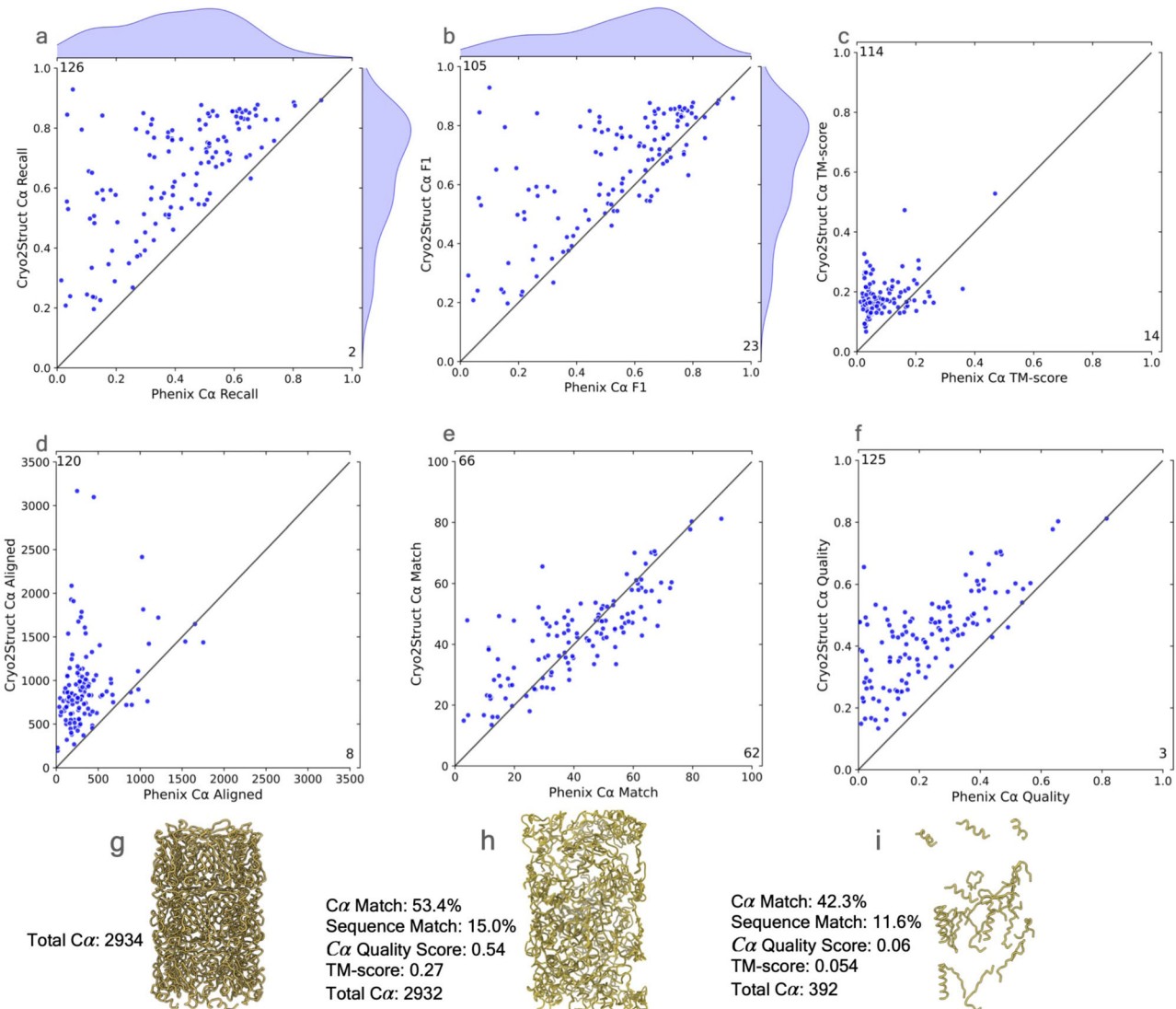

**Fig. 2 | The comparative analysis of atomic models built for 128 test cryo-EM maps by CryoStruct and Phenix in terms of six metrics.** In each panel of an evaluation metric, the score of the model built by CryoStruct for each map is plotted against that by Phenix for the same map. A dot above the 45 degree line indicates that CryoStruct has higher score than Phenix for the map. The number in the top-left corner represents the total number of maps on which CryoStruct has higher scores, while the number in the bottom-right corner denotes the total number of maps on which Phenix has higher scores. **a** The Cα recall of the atomic models of CryoStruct against Phenix; the recall is defined as the number of Cα atoms in the predicted model that are placed within 3 Å of the correct position in the corresponding known structure, divided by the total number of Cα atoms in the known structure. **b** The F1 score of Cα, which is the harmonic mean of precision and recall of Cα; it is a balanced measure quantifying a method's ability to make accurate Cα predictions while also capturing as many Cα atoms as possible. **c** The

TM-score of the atomic models normalized by the length of the known structure; the normalized TM-score is calculated by using US-align to align the atomic models with their corresponding known structures. **d** The length of aligned Cα atoms; it is calculated by using US-align to align the predicted model and the known structure. **e** The Cα match score of the atomic models; it is calculated by using Phenix.chain_comparison tool to compare them with the known structures. **f** The Cα quality score; it is the product of the Cα match score and the total number of predicted residues divided by the total number of residues in the experimental structure; the total number of predicted residues is calculated by Phenix.chain_comparison tool. **g** The true structure of EMD ID: 8767 (PDB ID: 5W5F); the map was released on 2017-08-16 with resolution of 3.4 Å. **h** The Cryo2Struct modeled structure and its scores. **i** The Phenix modeled structure and its scores. Source data are provided as a Source Data file.

score, the percentage of Cα atoms (residues) in the model that have corresponding residues within a 3 Å distance in the true structure. It also reports the sequence match score, i.e., the percentage of the matched residues that have the same amino acid type (identity) as their counterparts in the true structure. The models built by Cryo2Struct have an average Cα match score of 43%, higher than Phenix's 41.2%. The average sequence match score is 13.4% for both Cryo2Struct and Phenix. It is worth noting that the Cα match score measures the match precision of Cα atoms in a model without considering the Cα atom coverage of the model. For instance, a partial model may have a high Cα match score but can only cover a small portion of its

corresponding true structure. Because Cryo2Struct tends to build much more complete models than Phenix, their difference in terms of the Cα match score is less pronounced than in terms of the other metrics.

To remedy the shortcoming of the Cα match score calculated by the phenix.chain_comparison tool, we introduce a new Cα quality score considering both the Cα match precision and Cα coverage, which is the product of the Cα match score and the total number of predicted residues of a model obtained from the Phenix.chain_comparison tool divided by the number of the residues in the true structure. It is in the range [0, 1]. A higher score signifies a

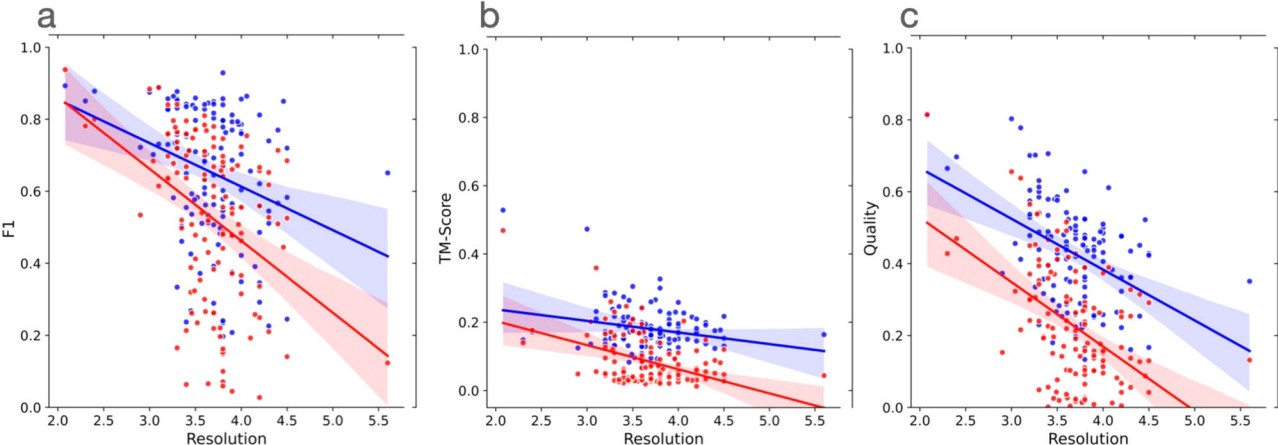

**Fig. 3 | The plots of the scores (F1 score, global normalized TM-score, and Cα quality score) of the models built by Cryo2Struct and Phenix against the resolution of the 128 cryo-EM density maps.** Blue dots denote Cryo2Struct constructed models and red dots the Phenix models. The solid lines depict linear regression lines, and the colored area represents a 95% confidence interval. The confidence interval is narrower (i.e., the linear estimation is more certain) in the resolution range [3–4.5 Å] where there are more data points. **a** F1 score against resolution. The equation of the regression line for Cryo2Struct (blue) is $y = -0.1209x + 1.0966$, while for Phenix (red), it is $y = -0.1998x + 1.2618$. The correlation between F1 score of Cryo2Struct and the resolution is −0.28, while for Phenix, it is −0.40. **b** The normalized global TM-score against resolution. The equation of the regression line for Cryo2Struct is $y = -0.0339x + 0.3057$, while for Phenix, it is $y = -0.0706x + 0.3447$. The correlation for Cryo2Struct is −0.24, while for Phenix, it is −0.43. **c** Cα quality score against resolution. The equation of the regression line for Cryo2Struct is $y = -14.1318x + 94.8512$, while for Phenix, it is $y = -17.9190x + 88.6207$. The correlation for Cryo2Struct is −0.43, while for Phenix it is −0.49. Source data are provided as a Source Data file.

more accurate and complete structural model. Fig. 2f compares the Cα quality scores of the structures modeled by Cryo2Struct and Phenix. The average Cα quality score for Cryo2Struct is 0.43, substantially >0.23 of Phenix. On 125 out of 128 maps, Cryo2Struct has a higher Cα quality score than Phenix. The result shows that Cryo2Struct is capable of building structural models with higher average coverage and Cα matching score than Phenix. Figure 2g–i illustrates such an example (EMD ID: 8767). The true structure for the map (Fig. 2g) has 2934 residues. The model built by Cryo2Struct (Fig. 2h) has 2932 residues, about 7.5 times 392 residues of the model built by Phenix (Fig. 2i) that is very fragmented, while the Cα match score and sequence match score of the former (i.e., 53.4% and 15%) are only 26–29% higher than 42.3% and 11.6% of the latter. In contrast, the Cα quality score of the Cryo2Struct constructed model is 0.54, 9 times 0.06 of the Phenix model, more accurately reflecting the difference in the quality of the two models.

Finally, we analyzed how the performance of the two methods changed with respect to the resolution of the cryo-EM density maps. Figure 3a–c plots the F1 scores, global normalized TM-scores, and Cα quality scores of the models built by the two methods against the resolution of the cryo-EM density maps measured in Angstrom (Å), respectively. In terms of each of the three scoring metric, as expected, the accuracy of models built by the two methods decreases as the value of the resolution of the cryo-EM density maps increases (i.e., the resolution gets worse). The linear regression line for CryoStruct models is above that for Phenix, indicating that for the maps of the same resolution, the average score of the models built by CryoStruct is higher than that of Phenix. Moreover, the gap between the two increases as the value of resolution gets larger. This indicates that the quality of the models built by Phenix decreases faster than CryoStruct as the resolution of the cryo-EM density maps gets worse, i.e., CryoStruct is more robust against (or less sensitive to) the change of the resolution of density maps than Phenix. This is reflected by the less steep negative slope of the regression line for Cryo2Struct than that of Phenix and the less negative correlation between the score of Cryo2Struct and the resolution of the cryo-EM density maps than Phenix's. For instance, the Pearson correlation coefficient between the F1 score of Cryo2Struct and the resolution is −0.28, weaker than −0.40

of Phenix. This observation is consistent in terms of all three metrics, indicating that Cryo2Struct generally builds better models from cryo-EM density maps than Phenix and therefore can be used to improve the quality of the models built from both the existing cryo-EM density maps in the Electron Microscopy Data Bank (EMDB) and the new ones to be generated.

### Evaluating Cryo2Struct on a large new dataset

We further evaluated the performance of Cryo2Struct on a large independent test dataset of 500 new maps with resolutions ranging from 1.9 Å to 4.0 Å. The average resolution of the density maps is 2.88 Å. These maps, released after April 2023, do not exist in the training and validation data in Cryo2StructData[24] that contains the cryo-EM density maps released before April 2023. The number of residues in the 500 maps ranges from 234 to 8828.

On the new dataset, the average recall, F1 score, global normalized TM-score, Cα quality score, Cα sequence match score, and Cα match score of Cryo2Struct are 70%, 70%, 0.22, 0.50, 20.1%, and 49.5%, respectively, higher than 65%, 66%, 0.2, 0.43, 13.4%, and 43% on the standard test dataset, suggesting that the average quality of the cryo-EM density maps in the new dataset is higher than the standard dataset, which is consistent with the fact that the new cryo-EM density maps have the average resolution of 2.88 Å better than the average resolution of 3.68 Å of the old density maps in the standard test dataset. The relatively high recall, F1 score, Cα quality score, and Cα match score show that Cryo2Struct performs very well in identifying individual Cα atoms, while the relatively lower global normalized TM-score and Cα sequence match score indicates it is still very challenging to build correct connected models that cover and match most regions of a large protein structure and its sequence.

Figure 4a–f illustrates the relationship between each of the six scores (the recall, F1 score, global normalized TM-score, Cα quality score, Cα sequence match score, and Cα match score) of the models and the resolution of the density maps. In terms of each metric, there is a negative relationship between the metric and the resolution, i.e., the quality of model decreases as the resolution value of cryo-EM density map increases (i.e., the resolution gets worse) as

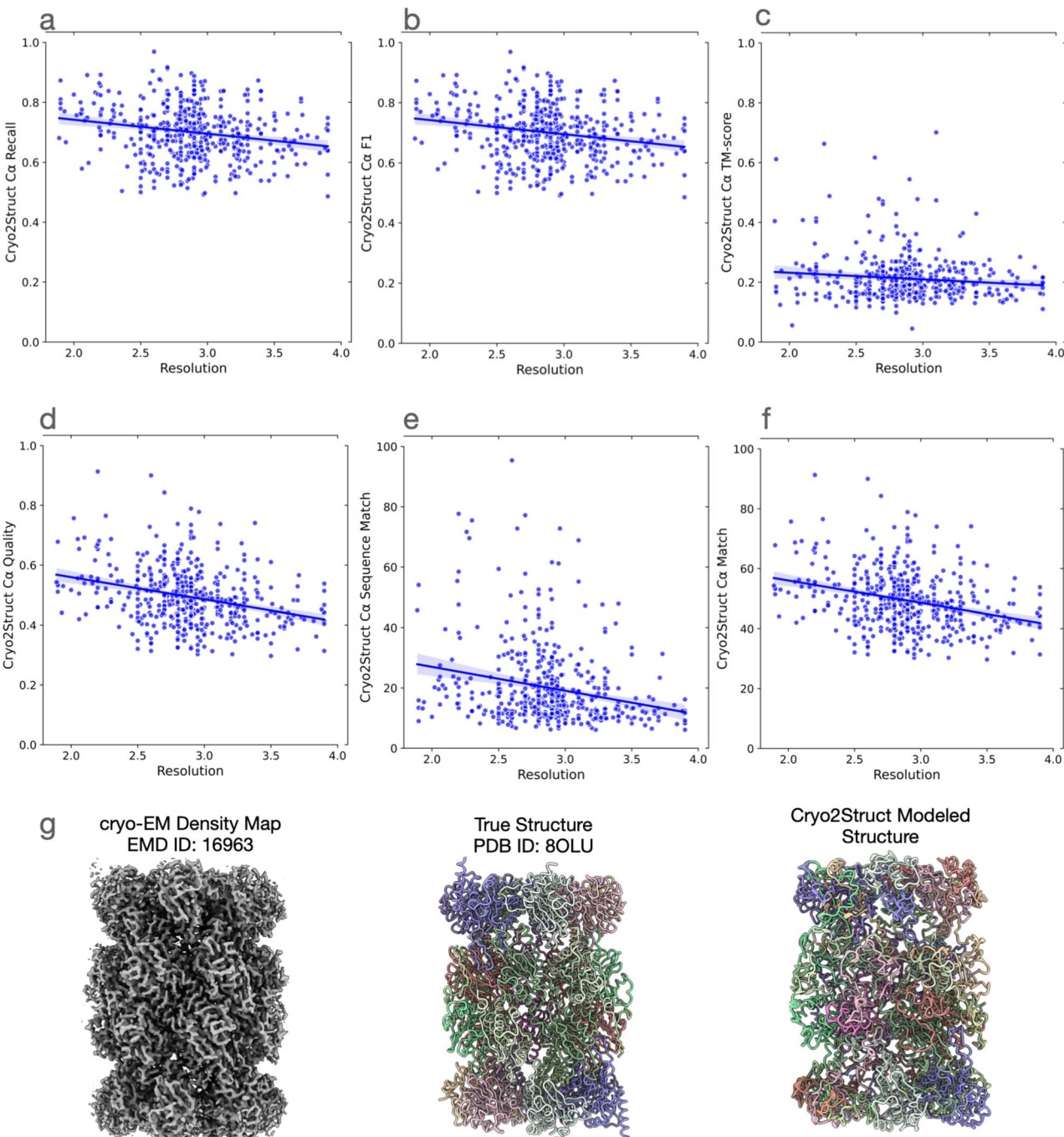

$C\alpha$ Quality Score: 0.73 - TM-score: 0.43 - $C\alpha$ Match: 72.8% - Sequence Match: 50.6% - F1: 90.5% - Total $C\alpha$: 6316

**Fig. 4 | The quality of atomic models built for 500 test cryo-EM maps.** The solid lines depict linear regression lines, and the colored area represents a 95% confidence interval. **a** The $C\alpha$ recall versus resolution; the regression equation: $-0.0466x + 0.8350$; Pearson's correlation: $-0.201$. **b** The F1 score versus resolution; the regression equation: $-0.0468x + 0.8357$; the correlation: $-0.202$. **c** The normalized TM-score versus resolution; the regression equation: $-0.0222x + 0.2762$; the correlation: $-0.11$. **d** The $C\alpha$ quality score versus resolution; the regression equation: $-0.0741x + 0.7080$; the correlation: $-0.298$. **e** The $C\alpha$ sequence match score versus resolution; the regression equation: $-7.9226x + 42.8422$; the correlation: $-0.234$. **f** The $C\alpha$ match score versus resolution; the regression equation: $-7.4408x + 70.8924$; the correlation: $-0.299$. **g** A modeling example. One on the left is the cryo-EM density map (EMD ID: 16963), in the middle is the true structure (PDB ID: 8OLU), and on the right is the model built by Cryo2Struct. The structure is a hetero 28-mer with a stoichiometry of A2B2C2D2E2F2G2H2I2J2K2L2M2N2 and a weight of 848.37 kDa. The total number of modeled $C\alpha$ atoms is 6316. Source data are provided as a Source Data file.

observed on the standard test dataset. The Pearson correlation coefficient (PCC) for the recall, F1 score, global normalized TM-score, $C\alpha$ quality score, $C\alpha$ sequence match, and $C\alpha$ match scores with respect to the resolution are $-0.201, -0.202, -0.11, -0.298, -0.234$, and $-0.299$, respectively, indicating that the negative relationship is rather weak and Cryo2Struct is robust against the

deterioration of the resolution of cryo-EM density maps. Fig 4g illustrates a high-quality model for a very large protein complex (EMD ID: 16963). The model has 6316 residues and high-quality scores ($C\alpha$ quality score = 0.73, TM-score = 0.43, $C\alpha$ match score = 72.8%, sequence match score = 50.6%, and F1 score = 90.5%). Furthermore, Fig. 5 shows several additional good modeling examples,

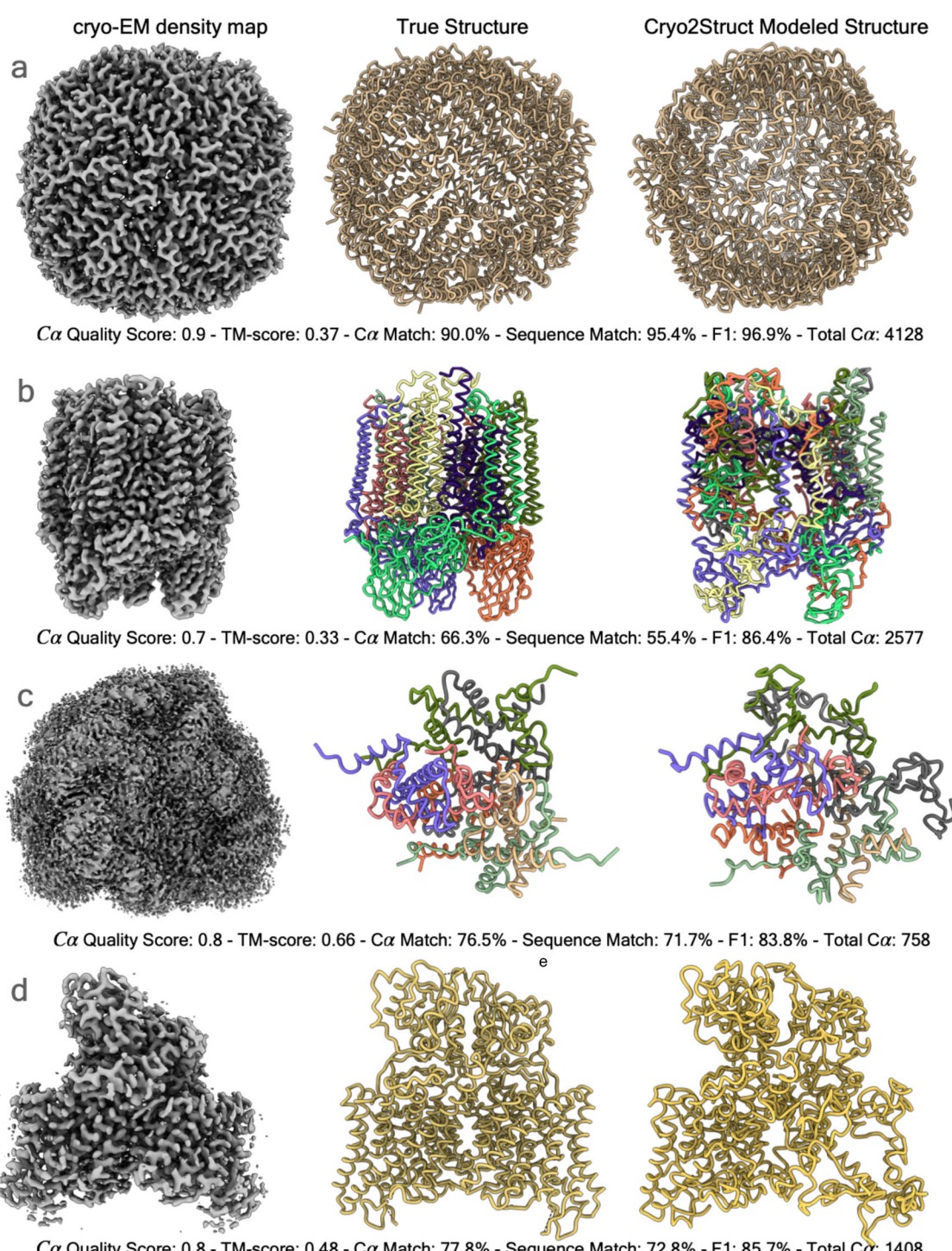

Fig. 5 | **The high-quality models built for four test cryo-EM maps.** In each panel from left to right are the cryo-EM density map, the true structure, and the model built by Cryo2Struct. The chains in both the true structure and the model are colored with distinct colors. The total Cα number shown in each panel is the total number of residues in a model. **a** The result for EMD ID: 17961 (PDB ID: 8PVC, released on 2023-11-29, and resolution of 2.6 Å). **b** The result for EMD ID: 17287 (PDB ID: 8OYI, released on 2023-11-08, and resolution of 2.2 Å). **c** The result for EMD ID: 37070 (PDB ID: 8KB5, released on 2023-10-18, and resolution of 2.26 Å). **d** The result for EMD ID: 35299 (PDB ID: 8IAB, released on 2023-08-02, resolution of 2.96 Å). Source data are provided as a Source Data file.

demonstrating that Cryo2Struct is capable of modeling some large structures with good overall accuracy.

Moreover, because only some regions of the models built by Cryo2Struct can be aligned with the true structures, we specifically analyzed the quality of the local regions of the models that can be aligned with the true structures by US-align in terms of aligned Cα length and RMSD (root mean squared distance) of the aligned regions.

Supplementary Figure S2 plots the RMSD of the aligned regions against their lengths for all the models built for the density maps in the new test dataset. The average length of the aligned regions of the models is 532.51 residues, accounting for 29% of the average length of true structures (i.e., 1837.43 residues). And the average RMSD of the aligned regions is 1.6 Å. The results show that Cryo2Struct can build a significant portion of the protein structures with very high accuracy

(low RMSD). And the RMSD decreases (i.e., the accuracy increases) with respect to the length of the aligned regions, according to the weak Pearson's correlation of −0.134 between the RMSD and the length of aligned regions. It is interesting to observe that Cryo2Struct can build high-accuracy models of large aligned regions up to thousands of residues.

Finally, we investigated how the global quality of the models changes with respect to the length (number of the residues) of the known structures (i.e., the size of the proteins) (supplementary Fig. S3). Unlike their similar relationship with the resolution of the cryo-EM density maps, the six metrics (recall, F1 score, global normalized TM-score, Cα quality score, Cα sequence match score, and Cα match score) exhibit different relationship with the size of the proteins. The Cα recall and F1 score have a weak positive correlation (i.e., 0.259 and 0.258 respectively) with the size of proteins indicating that it is slightly easier to recognize individual Cα atoms for larger protein structures, while there is a weak negative correlation (i.e., −0.214) between the global TM-score and the size of proteins indicating it is slightly more difficult to build accurate full-length models for larger proteins. And the correlation for Cα quality score, Cα sequence match score, and Cα match score with respect to the size of proteins is almost 0, indicating that these scores are largely independent of the size of proteins.

### Evaluating Cryo2Struct on highly sequence-dissimilar proteins

To investigate how well Cryo2Struct can generalize to proteins that are highly dissimilar to the proteins in the training and validation dataset, we used MMseqs2[25] to compare the proteins in the standard test dataset and the new test dataset with those in the training and validation dataset and removed any protein in each of them that contains one or more chains having more than 25% sequence identity with any chain of any protein in the training and validation dataset. The stringent 25% sequence identity is a threshold also utilized by DeepMainmast[12] in preparing a non-redundant test dataset. After the filtering, 22 out of 128 cryo-EM density maps in the standard test dataset are left to form a redundancy-reduced standard test dataset. The resolution of the density maps in the redundancy-reduced standard test dataset ranges 2.08–5.6 Å and has an average resolution of 3.72 Å and the number of residues in the maps ranges from 448 to 7440. Likewise, 169 out of 500 cryo-EM density maps in the new test dataset are left to form a redundancy-reduced new test dataset. The resolution of the density maps in the redundancy-reduced new test dataset ranges 1.93–3.9 Å and has an average resolution of 2.89 Å and the number of residues in the maps ranges from 234 to 7248.

Supplementary Figure S5 compares Cryo2Struct with Phenix on the redundancy-reduced standard test dataset in terms of three metrics: recall, F1, and quality score. Cryo2Struct performs better across the board, with 21 out of 22 structures having higher recall and quality scores and 16 out of 22 structures having higher F1 scores. The average recall score of Cryo2Struct is 67.8%, much higher than 40.7% of Phenix. Similarly, the F1 score for Cryo2Struct is 68%, higher than 51% of Phenix. The average quality score for Cryo2Struct is 0.51, much higher than 0.27 for Phenix.

Moreover, Cryo2Struct has an average sequence match of 18.2%, higher than 14.2% of Phenix. The Cα match for Cryo2Struct is 50.9%, higher than 48.5% of Phenix. The average length of Cryo2Struct predicted structures is 3224.5, close to the average length of the true structures (i.e. 3237.31) and more than double the average length of Phenix predicted structures (i.e., 1580.8). The aligned length for Cryo2Struct predicted structures is 955.8, much higher than 514.0 of Phenix. The global normalized TM-score for Cryo2Struct is 0.22, much higher than 0.13 of Phenix. The results demonstrate that Cryo2Struct substantially out perform Phenix on the test proteins that are highly dissimilar to the training and validation proteins.

Furthermore, Cryo2Struct's average recall, F1, global normalized TM-score, Cα quality score, Cα sequence match score, and Cα match score on the redundancy-reduced standard test dataset are 67.8%, 68%, 0.22, 0.51, 18.2%, and 51%, respectively, very similar to its scores of 65%, 66%, 0.2, 0.43, 13.4%, and 43% on the standard test dataset without the redundancy reduction, indicating that Cryo2Struct's performance is independent of sequence similarity and it generalizes well to test proteins that have little or no sequence similarity with the proteins in the training and validation dataset.

Supplementary Figure S6 plots the recall, F1, and quality scores of Cryo2Struct predicted structures against the resolution of the density maps in the redundancy-reduced new test dataset. Similarly, as on the new test dataset, the scores slightly trends down as the resolution gets worse. Cryo2Struct's average recall, F1, global normalized TM-score, Cα quality score, Cα sequence match score, and Cα match score on the redundancy-reduced new test dataset are 70.4%, 70.5%, 0.22, 0.51, 21.2%, and 50.5%, respectively, very similar to its scores of 70%, 70%, 0.22, 0.50, 20.1%, and 49.5% on the new test dataset without the redundancy reduction. This is same as observed on the redundancy-reduced standard test dataset and the standard test dataset, further confirming that Cryo2Struct's performance generalizes well to new proteins that are highly dissimilar to the proteins in the training and validation data. The decoupling of CryoStruct's performance and sequence similarity is probably because its transformer model only uses electron density values in cryo-EM density maps to predict Cα atoms and their amino acid types without using protein sequence information at all.

### Confidence scores by Cryo2Struct

Cryo2Struct provides a per-residue estimation of confidence within the range of [0, 1] for both Cα and amino acid type predictions, i.e., the chance (probability) that the predicted Cα or amino acid type is correct. Like the pLDDT scores that AlphaFold[16] assigns to predicted structures, the confidence scores reflect the degree of confidence Cryo2Struct has in predicted Cα atoms and their amino acid types, with higher scores indicating more reliable predictions, while lower scores suggest more uncertainty that warrants scrutiny of the predictions.

Specifically, the confidence score for a predicted Cα atom is estimated by one logistic regression classifier ($P(y=1|x) = \frac{1}{1+e^{-(\beta_0 + \beta_1 \cdot x)}}$; $\beta_0$ and $\beta_1$: weights to be optimized), which utilizes the probability of the Cα atom predicted by the deep learning model as input ($x$) to assess its probability of correctness ($P(y=1|x)$). Similarly, the confidence score for a predicted amino acid type is estimated by another logistic regression classifier using the emission probability of the amino acid type from the HMM assigned by the Viterbi algorithm, the probability of the Cα atom predicted by the deep learning model, and the one-hot encoding of the amino acid type as input ($x$) to predict the correctness of the amino acid type prediction.

To generate the binary labels (1: correct and 0: incorrect) to train the logistic regression classifiers, we utilized Phenix.chain_comparison to match a Cryo2Struct modeled structure with the corresponding true structure. We assigned a label of 1 to the Cα atoms in the Cryo2Struct modeled structure that have matching residues in the true structure, otherwise a label of 0. For the matched Cα atoms, we further assigned a label of 1 to the amino acid types matched with those in the true structure, and 0 otherwise.

We trained the two logistic regression classifiers on the 325 Cryo2Struct modeled structures of 325 targets in the new test dataset and tested them on the separate subset of 167 Cryo2Struct modeled structures that have less than 25% sequence identity with the Cryo2Struct training dataset, which is the same subset used in the section titled: Evaluating Cryo2Struct on highly sequence-dissimilar proteins.

To assess how well the confidence scores generated by the logistic regression can measure the quality of the 167 test structures built by Cryo2Struct, we correlated them with the true Cα and sequence match scores of the structures computed by phenix.chain_chain_comparison with respect to the true structures. The overall Cα confidence score for a Cryo2Struct modeled structure is the average of the confidence scores of all its Cα atoms, and the overall amino acid type confidence scores for a Cryo2Struct modeled structure is the average of the confidence scores of the amino acid types of all its residues. The Pearson's correlation coefficient between the Cα match score and the Cα confidence score for the 167 test structural models is 0.6, with a corresponding $p$-value of 3.06E-17. This correlation value suggests a strong positive linear relationship between the Cα match score and the Cα confidence score. The low p-value indicates that the observed correlation is statistically significant. Similarly, the correlation between the sequence match score and the overall amino acid confidence type score for the 167 test structural models is 0.7, with a $p$-value of 1.72E-24, confirming that the latter is a good indicator of the former.

Furthermore, there is a strong relationship between the per-residue Cα atom confidence scores and amino acid type confidence scores on the 167 test structural models built by Cryo2Struct. The correlation coefficient between them is 0.96, with a low p-value of 1.14E-92. Supplementary Figure S7 show an example plotting the confidence scores of Cα atoms against the confidence scores of amino acid types for a test protein, indicating a robust positive connection between the two kinds of confidence scores. Supplementary Figure S8 shows the two test examples of visualizing amino acid type confidence scores on top of the Cryo2Struct modeled structures using a color spectrum. Supplementary Figure S9 uses a detailed test example to demonstrate how different confidence scores can help users to identify high/local quality regions in the structural model in comparison with the true structure.

### Refinement of modeled structures

In the sections above, we performed a comparison between the Cryo2Struct modeled structures and the structures generated by the Phenix.map_to_model tool. Phenix.map_to_model employs an integrated procedure that combines various independent modeling methods with an extensive real-space refinement technique[13,26] to generate the structures. The models computed by Phenix.map_to_model benefit from the refinement through the Phenix.real_space_refine tool, which ensures their geometric integrity by resolving torsion angle outliers and rotamer outliers.

To investigate if the real-space refinement technique can further improve the Cryo2Struct modeled structures, we applied the same Phenix.real_space_refine tool employed by Phenix.map_to_model to refine the initial models built by Cryo2Struct. On the new test dataset, the average Cα match score of the refined models is 56%, 6.5% higher than that of the initial models, and the average RMSD of the refined models is 1.4 Å, 0.2 Å lower than that of the initial models. Similarly, on the standard test dataset, we observed an improvement in the average Cα match score by 8.9% along with a decrease of 0.2 Å in the RMSD, yielding an average Cα match score of 51.8% and an average RMSD of 1.62 Å for the refined models, respectively. This provides the compelling evidence that refining the initial models generated by Cryo2Struct further improves their quality by rectifying some geometrical issues.

## Discussion

De novo modeling of protein structure solely from density maps, without using structural templates, is an interesting and important issue because it establishes a lower bound on the amount of structural information that can be extracted from density maps. We developed Cryo2Struct, a de novo AI modeling method based on the transformer and HMM for building atomic protein structural models from medium- and high-resolution cryo-EM maps alone. The modeling process is fully automated, requiring no human intervention and no input from external tools. Cryo2Struct can rather accurately identify individual Cα atoms in density maps and is robust against the decrease of the resolution of density maps. Moreover, Cryo2Struct achieved substantially better performance than the most widely used de novo modeling method - Phenix in terms of multiple evaluation metrics including Cα recall, F1 score, global normalized TM-score, aligned Cα length, Cα match score, Cα sequence match score, and Cα quality score. In general, it can build much more accurate and more complete protein structures from cryo-EM density maps than Phenix, therefore advancing the state of the art of ab initio modeling of protein structures on cryo-EM density maps and providing a useful means for the community to build better protein structural models from both existing cryo-EM density maps and new ones to be generated to support biomedical research.

However, even though Cryo2Struct can identify most Cα atoms correctly with high F1-score and build high-accurate atomic models for some regions of large protein structures with very low RMSD, building high-accurate models covering most regions of large protein structures from density maps alone remains very challenging, reflected in low global TM-score and Cα sequence match score of the models. Obtaining high global TM-score and Cα sequence match score requires most if not all individual Cα atoms not only being correctly identified but also being correctly linked as peptide chains and assigned with correct amino acid types, which is combinatorially more challenging than predicting individual Cα atoms. A prediction error for only a few Cα atoms caused by missing or noise values in cryo-EM density maps that are very common may drastically lower the TM-score and Cα sequence score of the models because only when a long continuous stretch of chains are correctly predicted, the high TM-score and Cα sequence match score can be obtained. However, experimentally generating cryo-EM density maps that contain high-resolution density values covering every residue of a protein structure is still very challenging.

We envision that the global TM-score and Cα sequence match score of the structural models built from cryo-EM density maps can be further improved from the following aspects. The first is to develop more sophisticated and robust AI methods to predict protein atoms and their amino acid types with higher sensitivity and specificity from cryo-EM density maps to help build more accurate and complete protein chains. The second is to use additional inputs such as protein sequence information and AlphaFold-predicted protein structures to complement missing information in cryo-EM density maps to obtain more accurate and complete predictions. The third is to leverage the symmetry of multiple chain in protein complexes to more accurately predict Cα atoms and amino acid types and align protein sequences with the HMMs. The fourth is to generate more accurate and complete cryo-EM density maps in the first place for the AI methods to use, which is being done by the community and would automatically improve the performance of Cryo2Struct as seen on the new test dataset in this work.

In the future, we plan to further expand Cryo2Struct to integrate cryo-EM density maps, protein sequences, and AlphaFold-predicted structures with deep learning together to build more accurate and complete protein structures. As more and more high-quality cryo-EM maps are being deposited in EMDB[9], such tools for automatically modeling atomic structure from them can enable scientists to better leverage this valuable resource to advance biomedical research.

## Methods
### Structure modeling process
As illustrated in Fig. 1, Cryo2Struct tackles the problem of building 3D atomic structural models from 3D cryo-EM density map in the following three main steps:

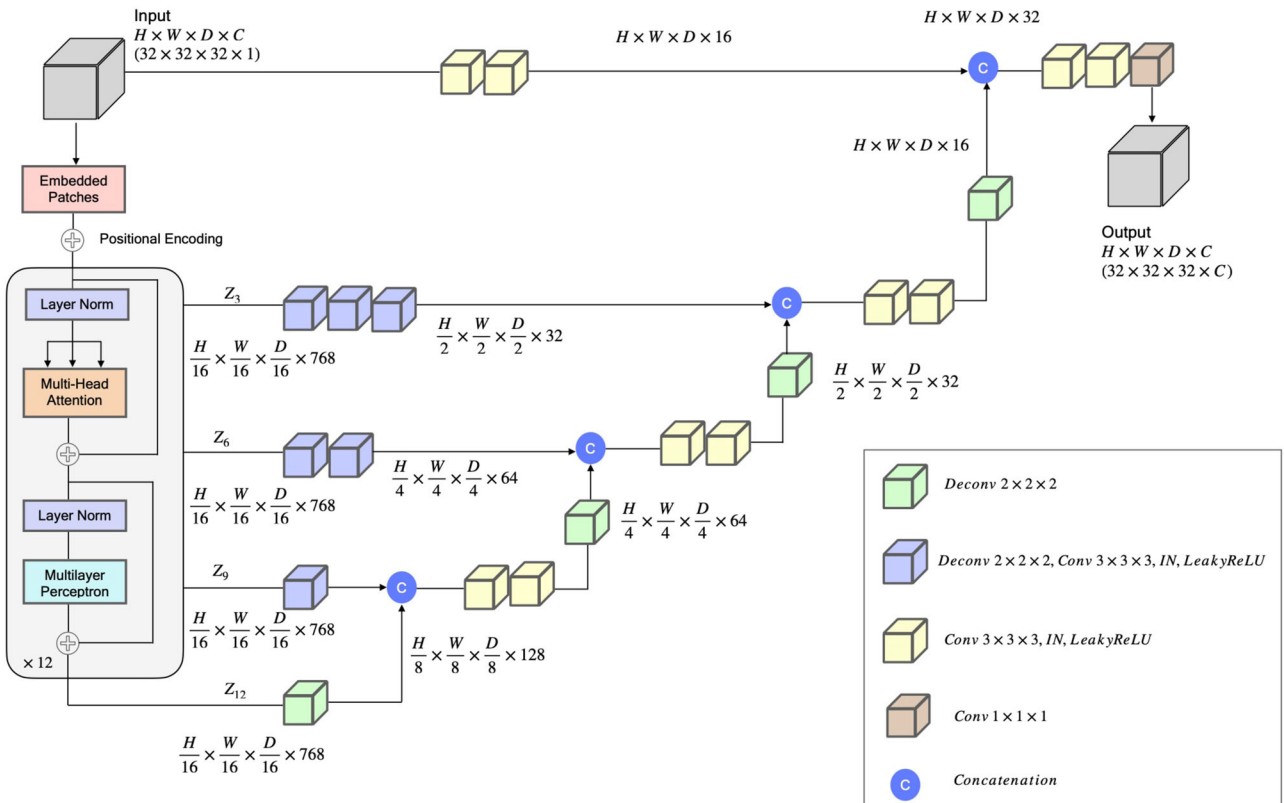

**Fig. 6 | The Deep Learning architecture for backbone atom and amino acid type classification.** The network takes a $32 \times 32 \times 32$ sub-grid of cryo-EM density map as an input with one channel representing the density value of voxels. The input is divided into a series of patches. The patches are projected into an embedding space by a 3D convolution layer, and then is added with a positional encoding. The patches are then processed by an encoder, comprising 12 identical blocks each with a normalization layer, a multi-head self-attention layer, a normalization layer, and a multi-layer perceptron (MLP). The encoded features of blocks 3, 6, 9 and 12 denoted as ($z_3, z_6, z_9, z_{12}$) and the original input are integrated into the decoders via skip connections in a U-Net fashion, each of which includes convolution and deconvolution layers with instance normalization (IN), Leaky ReLU activation, and feature concatenation. The last hidden features are used by a $1 \times 1 \times 1$ convolution layer to generate the final 3D sub-grid output of the same size as the input, i.e., $32 \times 32 \times 32$, with (C) output channels (i.e., 4 for the backbone atom type classification (C$\alpha$, N, C, and the absence of an atom) and 21 for the amino acid type classification (20 standard amino acids and no/unknown amino acid). The amino acid-type classification model has 92.281893 million parameters, whereas the atom type classification model has 92.281604 million parameters.

1. Predict C$\alpha$ voxels and their amino acid types in the cryo-EM density map of a protein using a deep learning method based on transformer.
2. Construct a HMM model ($\lambda$) with predicted C$\alpha$ voxels as hidden states and with emission and transition probability parameters set according to their predicted probabilities and their pairwise distance.
3. Align the amino acid sequence (i.e., $O = O_1, O_2, O_3...O_T$, where $O_t \in V$; $V$: the set of 20 standard amino acids) with the HMM model $\lambda$ to find the most likely C$\alpha$ state sequence (path) ($X = x_1, x_2, x_3, ...$ $x_T$ where $x_t \in S$; $S$: the set of C$\alpha$ hidden states) of generating the sequence to form the backbone structure of the protein.

**Predicting C$\alpha$ voxels and amino acid types**

We designed a transformer-based model (Fig. 6), inspired by U-Net Transformers (UNETR)[27], for voxel classification in cryo-EM density maps. The model follows the contracting-expanding pattern of U-Net[28], utilizing a series of transformer-based encoders to extract features at multiple layers. The features extracted from different layers are utilized by a CNN-based decoder using skip connections to classify the voxels into different classes. One model is trained to classify voxels into four different classes (C$\alpha$, C, N, and the absence of an atom) (atom type classification). Another model is trained to classify voxels into 21 classes, representing 20 amino acid types and an absent or unknown amino acid type.

**Deep learning architecture.** The deep learning model (Fig. 6) takes in an input sub-grid of cryo-EM density map represented as a 4D tensor with dimensions $H \times W \times D \times C$, where $H$ is the height, $W$ is the width, $D$ is the depth, and $C$ is the number of channels (C=1 for the input), denoted as $x \in \mathbb{R}^{H \times W \times D \times C}$. $x$ is then divided into a series of flattened, uniform non-overlapping patches ($x_v \in \mathbb{R}^{N \times (P^3 \cdot C)}$), where $P$ denotes the patch dimensions and $N = (H \times W \times D)/P^3$ is the number of the patches. The series of the patches are projected by a 3D convolution layer into a $K$-dimensional embedding space. A 1D learnable positional encoding $\mathbf{E}_{pos} \in \mathbb{R}^{N \times K}$ is then added to the projected patches, which subsequently serve as the input to the transformer encoder. Here, $P$ is set to 16 and the embedding dimension ($K$) to 768.

Cryo2Struct uses an encoder of 12 blocks[17] each consisting of a normalization layer[29], a multi-head attention layer, a normalization layer and a multi-layer perceptron to generate features for the input series of patches. The features from four different blocks: $z_i$ (i.e., $i \in \{3, 6, 9, 12\}$), with size $\frac{H \times W \times D}{P^3} \times K$ are reshaped into $\frac{H}{P} \times \frac{W}{P} \times \frac{D}{P} \times K$, respectively. The features of the four blocks and the original input are processed by deconvolution and/or convolution layers and concatenated together in a U-Net fashion step by step to generate the final feature tensor of the same dimension as the original input (see Fig. 6 for details), which is used by a $1 \times 1 \times 1$ convolution layer to classify each voxel.

**Training and validation.** We used the Cryo2StructData[24] dataset, which includes maps with the resolution in the range [1.0–4.0 Å], to

train and validate the two transformer models. The cryo-EM density maps in the dataset were released till 27 March 2023. The dataset is split according to a 90% to 10% ratio into the training and validation datasets. The total dataset has 7392 cryo-EM density maps. The training dataset and validation dataset has 6652, and 740 cryo-EM density maps, respectively. The atom types and amino types of the voxels in the density maps are labeled.

The training was performed on sub-grids (dimension: $32 \times 32 \times 32$) of the density maps, utilizing a batch size of 720, the NADAM optimizer[30] with a learning rate of 1e-4, and a dropout rate of 0.1. We used a distributed data parallel (DDP) technique to train the models on 24 compute nodes each equipped with 6 NVIDIA V100 32GB-memory GPUs in the Summit supercomputer[31].

The deep learning models were trained with the weighted cross entropy loss function described in Eq. (1) to handle the class imbalance problem.

$$\mathcal{L}(x,y) = -\frac{1}{N}\sum_{n=1}^{N}\sum_{c=1}^{C} w_c \cdot y_{n,c} \cdot$$
$$\log\left(\frac{\exp(x_{n,c})}{\sum_{i=1}^{C}\exp(x_{n,i})}\right) \quad (1)$$

where, $\mathcal{L}(x,y)$ represents the weighted cross-entropy loss. $N$ is the number of samples in the minibatch. $C$ is the number of classes. $w_c$ is the weight for class $c$ computed using Formula (2). $x_{n,c}$ is the logit for class $c$ in sample $n$, and $y_{n,c}$ is a binary indicator (0 or 1) of whether class $c$ is the correct classification for sample $n$. $\omega_c$ in Formula (2) represents the weight assigned to class $c$, $n_c$ is the number of samples in class $c$, and $\sum_{k=0}^{\text{classes}} n_k$ is the total number of samples across all classes.

$$\omega_c = 1 - \frac{n_c}{\sum_{k=0}^{\text{classes}} n_k} \quad (2)$$

Throughout the training process, we monitored both training and validation loss along with the F1 score, known for its effectiveness in handling class-imbalanced data as it represents the harmonic mean of precision and recall. We implemented and trained the deep learning models using PyTorch Lightning[32], version 1.7.3. The evaluation metrics (F1, Recall, and Precision) were computed using TorchMetrics[33], version 0.9.3. We tracked the model's performance on both training and validation data using the Weights and Biases tool. If the validation loss did not improve for five consecutive epochs, we reduced the learning rate by a factor of 0.1. We saved the top 5 trained models with lowest validation loss during the training and selected the model with the highest F1 score on the validation dataset as the final trained model.

**Cα voxel clustering.** When applying the trained transformer to a density map to predict Cα voxels, it is common that multiple spatially close voxels corresponding to the same Cα atom are predicted as Cα atoms. To remove redundancy, Cryo2Struct employs a clustering strategy to group predicted Cα voxels within a 2 Å radius into clusters. The average Cα probability and the amino acid type probability of Cα voxels in each cluster are computed. The centrally located Cα voxel in each cluster and the average probabilities of the cluster are used to represent the Cα atom of the cluster, while the other Cα voxels in the same cluster are removed.

**Connecting Cα atoms into protein chains and assigning amino acids to Cα atoms**

Connecting predicted Cα voxels into chains and accurately assigning their amino acid types is a challenging task. We designed an innovative Hidden Markov Model (HMM) whose hidden states represent predicted Cα voxels to accomplish it seamlessly in a single step, which is used by a customized Viterbi algorithm to align the sequence of a target protein with the HMM. The hidden states (Cα voxels) aligned with the sequence are joined together to form the backbone of the protein, in which the amino acid type of each Cα voxel is set to the type of the amino acid aligned with it.

Cα voxels with a probability higher than 0.4 are selected as the hidden states for the HMM. The HMM uses $K$ hidden states to represent predicted $K$ Cα voxels. Let's denote individual Cα hidden states in the HMM as $S = S_1, S_2, S_3, ..., S_K$ and individual symbols (amino acid types) as $V = V_1, V_2, V_3, ..., V_N$, where $N$ is equal to the number of standard amino acids (i.e., 20) generated from the hidden states. The hidden states in the HMM are fully connected, where there is a direct transition from any state to any other state, as depicted in supplementary Fig. S4a. The transition probabilities between Cα hidden states are stored in the transition matrix, denoted as $\gamma$ with a size of $K \times K$. The emission probabilities of generating observation symbols from the hidden states are stored in the emission matrix, denoted as $\delta$, with a size of $K \times N$. The initial state distribution is denoted as $\Pi = <\pi_1, \pi_2, \pi_3, ..., \pi_K>$, where $\pi_i$ is the probability that the HMM starts from state $i$. A hidden path may start from and end at any state. We use a compact notation, $\lambda = (\gamma, \delta, \Pi)$, to represent the HMM.

**Hidden Markov Model construction.** The transition probability matrix ($\gamma$) is constructed based on the distance between two predicted Cα states (voxels) in the 3D space, calculated from their coordinates using Formula (3). The distance $x$ is converted into a probability using the modified Gaussian probability density function (PDF) in Equation (4) ($f(x)$), with a mean ($\mu$) of 3.8047 Å and a standard deviation ($\sigma$) of 0.036 Å. Both $\mu$ and $\sigma$ were estimated from the distances between two adjacent Cα atoms in the true protein structures in the training dataset. Additionally, we introduce a fine-tune able scaling factor ($\Lambda$) that multiplies with ($\sigma$) to make the model adjustable. We set ($\Lambda$) to 10. The transition probabilities from one state to all other states are normalized by dividing each of them by their sum.

$$\sqrt{(x_2 - x_1)^2 + (y_2 - y_1)^2 + (z_2 - z_1)^2} \quad (3)$$

$$f(x) = \frac{1}{\Lambda\sigma\sqrt{2\pi}} e^{-(x-\mu)^2/2(\Lambda\sigma)^2} \quad (4)$$

The emission probability matrix ($\delta$) for each Cα state (voxel) is calculated from both its predicted amino acid type probability and the background (prior) probability of 20 amino acids in the nature. Specifically, the geometric mean of the two is calculated as $\sqrt{a \times b}$, where $a$ corresponds to the predicted probability for each amino acid type, and $b$ represents the background frequency of the amino acid type, as shown in supplementary Fig. S10, that was precomputed from the true protein structures in the training dataset. The geometric means for 20 amino acid types are normalized by their sum as their final emission probability. An example of emission matrix is shown in supplementary Fig. S4b.

The initial probability for a Cα state ($\pi_i$) is the probability that it generates the first amino acid of the protein sequence normalized by the sum of these probabilities of all the Cα states.

**Aligning protein sequence with HMM using a customized Viterbi Algorithm.** The customized Viterbi algorithm is used to find the most likely path in the HMM to generate a protein sequence with the maximum probability. The only difference between the customized Viterbi algorithm and the standard Viterbi algorithm is that the former allows a hidden state to occur at most once in the aligned hidden state path, while the latter does not have such a restriction. The restriction is

needed because one hidden state denoting a C$\alpha$ voxel can only be aligned to (occupied by) one amino acid in a protein sequence. The details of the algorithm is depicted in Algorithm 1, generating a path $X = x_1, x_2, x_3, ..., x_T$, which is a sequence of states $x_t \in S$ aligned with a protein sequence (the observation $O$). For a multi-chain protein complex, the sequence of each chain is aligned with the HMM one by one. Once a chain is aligned, the states in the hidden path aligned with it are removed from the HMM before another chain is aligned. In the alignment process, it is ensured that any C$\alpha$ state occurs at most once in one hidden state path. One distinct strength of this HMM-based alignment approach is that every amino acid of the protein is assigned to a C$\alpha$ position as long as the number of the predicted C$\alpha$ voxels is greater than or equal to the number of the amino acids of the protein, which is usually the case when the 0.4 probability threshold is used to select predicted C$\alpha$ atoms to construct the HMM. This is the reason that Cryo2Struct builds very complete structural models from density maps.

**Algorithm 1. The customized Viterbi algorithm**

1: **function** ModifiedViterbi ($O, S, \Pi, \gamma, \delta$): $X$
2:   for each state $i = 1, 2, ..., K$
3:     $T_1[i,1] \leftarrow \Pi_i \cdot \delta_{i,O_1}$
4:     $T_2[i,1] \leftarrow 0$
5:   **end for**
6:   $visited\_states \leftarrow \{\}$
7:   **for** each observation $j = 2, 3, ..., T$
   **do**
8:       **for** each state $i = 1, 2, ..., K$ **do**
9:         **if** $i \notin visited\_states$ **then**
10:            path $\leftarrow T_1[i, j-1] \cdot \gamma_i \cdot \delta_{i,O_j}$
11:            $T_1[i,j] \leftarrow \max_k(\text{path})$
12:            $T_2[i,j] \leftarrow \arg\max_k(\text{path})$
13:         **end if**
14:       **end for**
15:       $z_j \leftarrow \arg\max_k(T_1[:, j])$
16:       $visited\_states \leftarrow visited\_states \cup \{z_j\}$
17:   **end for**
18:   $z_T \leftarrow \arg\max_k(T_1[:, T])$
19:   $x_T \leftarrow S_{z_T}$
20:   **for** $j = T-1, T-2, ..., 1$ **do**
21:     $z_{j-1} \leftarrow T_2[z_j, j]$
22:     $x_{j-1} \leftarrow S_{z_{j-1}}$
23:   **end for**
24:     **return** $X$
25: **end function**

The customized Viterbi algorithm is a dynamic programming algorithm implemented in C++ to achieve high computational efficiency. The source code is compiled with a high level of optimization and is provided as a shared library, which is then linked with the Python program of constructing the HMM.

### Inference and testing

After Cryo2Struct was trained and validated, it was blindly tested on a standard test dataset of 128 density maps and a large new dataset of 500 density maps. For each test map, the Cryo2Struct inference process consisting of the deep learning prediction and the HMM alignment was executed on compute nodes each with a 40GB GPU, 150 GB RAM, and 64 CPU cores. The deep learning prediction was carried out on the GPU, whereas the HMM alignment was executed on the CPU cores. The model building for the largest map (EMD ID: 40492 with resolution 2.9 Å), involving 8828 modeled residues, was completed in 9 hours on a compute node, while it took only 2.90 minutes to build a model for the smallest map (EMD ID: 36426 with resolution 3.3 Å) with 234 residues.

### Reporting summary

Further information on research design is available in the Nature Portfolio Reporting Summary linked to this article.

## Data availability

The dataset used to train and validate Cryo2Struct (Cryo2StructData) is available on the Harvard Dataverse[24], and the description of the data preparation and labeling process can be found in[21]. The detailed information about the test datasets including the EMD IDs of the density maps and the evaluation scores are provided in two Excel files (`Standard_test_data.xlsx` for the standard test dataset and `Cryo2Struct_test_data.xlsx` for the new test dataset) available at https://doi.org/10.7910/DVN/GQCTTD, and the true structures and the structural models built by Cryo2Struct and Phenix for the test density maps are also available at the same website. The two Excel files (`Standard_test_data.xlsx` for the standard test dataset and `Cryo2Struct_test_data.xlsx`) for the new test dataset) are also available in source data file. Additionally, two Excel files (`Standard_test_data_low_sim.xlsx` for the redundancy-reduced standard test dataset and `Cryo2Struct_test_data_low_sim.xlsx` for the redundancy-reduced new test dataset) are available in source data file and on Harvard Dataverse, accessible at 10.7910/DVN/GQCTTD. Source data are provided with this paper.

## Code availability

The source code for Cryo2Struct is available in the GitHub repository: https://github.com/jianlin-cheng/Cryo2Struct. This repository also includes instructions on running Cryo2Struct on cryo-EM maps to generate 3D atomic protein structures. Furthermore, to keep the codes of Cryo2Struct permanent, we published all code and instructions required to reproduce the results on Zenodo, an online research sharing platform with a permanent Digital Object Identifier number[34].

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

## Acknowledgements

This work was supported in part by an NIH grant (R01GM146340) to JC. The computation for this work was performed on the high performance computing infrastructure provided by Research Computing Support Services at the University of Missouri, Columbia MO. This research also used the computing resource of the Oak Ridge Leadership Computing Facility, which is a DOE Office of Science User Facility supported under Contract DE-AC05-00OR22725.

## Author contributions

J.C. conceived the project. J.C. and N.G. designed the method and experiment. N.G. prepared the datasets, implemented the method, carried out the experiments, and collected the results. N.G. and J.C. analyzed the results. N.G. and J.C. wrote the manuscript.

## Competing interests

The authors declare no competing interests.
