## [Peer Review File · Nature Communications]

Reviewers' Comments:

Reviewer #1:

Remarks to the Author:

I have carefully reviewed the manuscript detailing the use of Cryo2Struct, a novel approach for de novo atomic protein structure modeling from cryo-electron microscopy (cryo-EM) density maps. Cryo2Struct employs a 3D transformer deep learning architecture alongside a Hidden Markov Model to tackle the significant challenge of constructing accurate 3D atomic structures of protein complexes in the absence of precise templates. While the approach is innovative and appears to demonstrate improved performance over Phenix.map_to_model, there are several critical issues that must be thoroughly addressed prior to consideration for publication. My concerns are as follows:

Major concerns:

1. The deep learning model of the proposed method was trained on a large dataset from Cryo2StructData including 6652 training and 740 validation density maps, and were tested on two test sets of respectively 128 and 500 maps. It appears that there was no control for similarity among the training, validation, and testing sets in the development of the proposed method. For deep learning models, it's crucial to ensure that the datasets used for training, validation, and testing are independent of each other. I recommend that the authors consider using protein sequence identity as a metric for similarity.
2. In Fig.4g and Fig.5b-d, it seems that there are a lot of misconnections in the structure model built by the proposed method. I suggest the authors discuss more detailly about the possible reasons for the failure/success of the proposed method.

Minor concerns:

1. In Abstract, the authors claimed that "... it built more accurate models than on the standard dataset". I cannot figure out there's any necessity to compare the evaluation results on two different test sets.
2. In Introduction, the authors claimed that "Cryo2Struct first uses a Transformer-based deep learning model with an attention mechanism for capturing long range atom-atom interactions ...". This claim may be not appropriate here.
3. I don't think TM-score is a good metric for the proposed method.

Reviewer #2:

Remarks to the Author:

This paper presents a de novo method, Cryo2Struct, to predict atomic structure directly from a cryo-electron density map. Although templates are available for many proteins, de novo methods are essential in solving structures, since they provide genuine observation of the molecule directly from the density data. This method is built on the use of a U-net transformer to identify C-alpha atoms from the density map and a HMM to relate them to protein sequences of multiple chains in the map. The method was tested using a large data set of 128 cryo-EM maps and another set of 500 recently-deposited density maps of 1.9-4.0 angstrom resolution. Results show that it is able to build more accurate models with more coverage of the structures than Phenix. The results presented in this paper are significant, since they show a milestone in automatic prediction of atomic structures directly from a density map.

The main contributions:

- The method is an automatic method for de novo modeling of cryo-EM density maps, unlike most modeling tools that rely on input of prior knowledge. Even though it is challenging to solve structures automatically from a cryo-EM density map that often contain multiple chains, this paper

shows the potential of an automatic method.

- The use of HMM to relate C-alpha atom positions to the amino acids on multiple sequences is novel. One of the existing methods for this problem is the use of travel sales man algorithm with predetermined heuristics.
- Using a large data set, results show that Cryo2Struct has significant improvement in both accuracy and completeness of the models than for Phenix's map_to_model method to produce initial models. Since Phenix is a widely used tool, having a more accurate initial model, such as the one produced through Cryo2Struct, can potentially improve downstream refinement steps.

I have the following questions/comments.

- The C-alpha atoms appear to be detected quite accurately, but the TM scores are low, presumably due to the alignment step. Does the HMM method consider the symmetry among multiple chains in the density map/structure? How is it different from how Phenix handles the symmetry?
- The introduction mentioned a few more recent deep learning methods for this problem, but the comparison was only conducted for Phenix. Can authors clarify why it is not compared with those?

Reviewer #3:

Remarks to the Author:

In this work, the authors develop a tool for matching EM density maps to a sequence + alpha-carbon trace of the protein(s). The approach uses a transformer to identify voxels associated with the alpha carbons, and then an HMM to match sequences to the trace. The approach is quite effective and outperforms ab initio methods such as the standard Phenix platform. The tool is well described and the code made accessible both on github and codeocean for those who wish to use and/or extend the current tool.

The HMM approach determines the probability distribution of amino acids at each given Calpha based on the distribution of neighbors. This information could be converted into a confidence score - similar to the PLDDT scores used in other tools which is based on the match of a prediction to a training set of environments. Such a calculation would allow the determination of local confidence in prediction, in addition to the protein-wide metrics reported.

Response to Reviewer Comments

We would like to thank the reviewers for the insightful suggestions and comments to help us improve the manuscript. Below are our point-by-point responses to address all the concerns raised by the reviewers. All the changes made in the main manuscript are highlighted in red.

Reviewer #1

I have carefully reviewed the manuscript detailing the use of Cryo2Struct, a novel approach for de novo atomic protein structure modeling from cryo-electron microscopy (cryo-EM) density maps. Cryo2Struct employs a 3D transformer deep learning architecture alongside a Hidden Markov Model to tackle the significant challenge of constructing accurate 3D atomic structures of protein complexes in the absence of precise templates. While the approach is innovative and appears to demonstrate improved performance over Phenix.map_to_model, there are several critical issues that must be thoroughly addressed prior to consideration for publication. My concerns are as follows:

Major concerns:

1. The deep learning model of the proposed method was trained on a large dataset from Cryo2StructData including 6652 training and 740 validation density maps, and were tested on two test sets of respectively 128 and 500 maps. It appears that there was no control for similarity among the training, validation, and testing sets in the development of the proposed method. For deep learning models, it's crucial to ensure that the datasets used for training, validation, and testing are independent of each other. I recommend that the authors consider using protein sequence identity as a metric for similarity.

Response:

Thank you for the great comments. According to your suggestions, we performed new experiments to investigate how well Cryo2Struct can generalize to proteins that are highly dissimilar to the proteins in the training and validation dataset. We used MMseqs2 to compare the proteins in the standard test dataset and the new test dataset with those in the training and validation dataset and removed any protein in each of them that contains one or more chains having more than 25% sequence identity with any chain of any protein in the training and validation dataset. The stringent 25% sequence identity is a threshold also utilized by DeepMainmast in preparing a non-redundant test dataset. After the filtering, 22 out of 128 cryo-EM density maps in the standard test dataset are left to form *a redundancy-reduced standard test*

dataset. Likewise, 169 out of 500 cryo-EM density maps in the new test dataset are left to form a *redundancy-reduced new test dataset*.

We then compared Cryo2Struct with Phenix on the redundancy reduced standard test dataset. Cryo2Struct still performs substantially better than Phenix on the redundancy reduced standard test dataset in terms of all the metrics. Moreover, Cryo2Struct's performance on the redundancy reduced standard test dataset is very similar to its performance on the original standard test dataset without redundancy reduction, indicating that Cryo2Struct's performance is independent of sequence similarity and it generalizes well to test proteins that have no or little sequence similarity with the proteins in the training and validation dataset.

Moreover, we evaluated Cryo2Struct's performance on the redundancy reduced new test dataset, which is very similar to its performance on the new test dataset without redundancy reduction, further confirming that Cryo2Struct's performance generalizes well to new proteins that are highly dissimilar to the proteins in the training and validation data. The decoupling of Cryo2Struct's performance and sequence similarity is probably because its transformer model only uses electron density values in cryo-EM density maps to predict C α atoms and their amino acid types without using protein sequence information at all.

The detailed new results above and their analysis have been added into a new subsection "*Evaluating Cryo2Struct on highly dissimilar structures*". Two new figures (Figure Extended Data 5 and Figure Extended Data 6) visualizing the results were also added, which are discussed in the subsection.

2. In Fig.4g and Fig.5b-d, it seems that there are a lot of misconnections in the structure model built by the proposed method. I suggest the authors discuss more details about the possible reasons for the failure/success of the proposed method.

Response:

Thank you for your great suggestions. We have added a new subsection titled "*Confidence Scores by Cryo2Struct*" to the Results section to analyze the reasons of the failure and success of Cryo2Struct. In this subsection, we use the probability of C α atoms and amino acid type predicted by Cryo2Struct as the residue-wide confidence score of the predictions. Like the pLDDT scores that AlphaFold assigns to predicted structures, the confidence scores reflect the degree of confidence Cryo2Struct has in predicted C α atoms and their amino acid types, with higher scores indicating more reliable predictions, while lower scores suggest more uncertainty that warrants scrutiny of the predictions.

We added four new figures (Fig. Extended Data 7, Fig. Extended Data 8, Fig. Extended Data 9, and Extended Data 10) to analyze the residue-wide confidence scores for the four modeled structures depicted in Figures 5.b, 5.c, 5.d, and 4.g that contain a lot of misconnections. Across these four figures, the average confidence scores for predicted C α atoms range from 0.67 to 0.71, much higher than the average confidence scores of predicted amino acid types ranging from 0.16 to 0.2, indicating that predicting amino acid types is more challenging than predicting C α atoms. The more confident C α predictions facilitate aligning protein sequences with the HMM to build

more accurate structures, but the less confident amino acid type predictions can mislead the alignment process, resulting in incorrect connections of some C α atoms as shown in Figures 5.b, 5.c, 5.d. and 4.g. The prediction confidence is likely related to the local resolution of density maps, i.e., low-resolution regions in density maps may have low confident predictions, leading to less accurately modeled structures for the regions.

Minor concerns:

1. In Abstract, the authors claimed that “..., it built more accurate models than on the standard dataset”. I cannot figure out there’s any necessity to compare the evaluation results on two different test sets.

Response:

Thank you for pointing out this issue. We have revised the related sentence in the abstract to remove this comparison. The new sentences are “*its performance of building atomic structural models is rather robust against changes in the resolution of density maps and the size of protein structures. Cryo2Struct’s performance also generalizes well to test proteins that have little or no sequence similarity with the training and validation proteins*”.

2. In Introduction, the authors claimed that “Cryo2Struct first uses a Transformer-based deep learning model with an attention mechanism for capturing long range atom-atom interactions ...”. This claim may be not appropriate here.

Response:

Thank you for raising the issue. We agree. We have restructured the sentence in the revised manuscript to remove the claim as follows: “*Cryo2Struct first uses a Transformer-based deep learning model with an attention mechanism to identify atoms and their amino acid types in cryo-EM density maps.*”

3. I don’t think TM-score is a good metric for the proposed method.

Response:

Thank you for the insightful feedback. Indeed, the average TM-score of the protein structures reconstructed from cryo-EM maps appears to be low in comparison with most other evaluation metrics and likely underestimates the performance of model building. In the revised manuscript, we opt to keep the TM-score evaluation results to provide a comprehensive analysis of our method's performance. While acknowledging that TM-score is not a good metric in all scenarios, we believe it might offer some useful insights into the similarity between modeled structures and true structures and could complement other evaluation measures to some degree.

Reviewer #2

This paper presents a de novo method, Cryo2Struct, to predict atomic structure directly from a cryo-electron density map. Although templates are available for many proteins, de novo methods are essential in solving structures, since they provide genuine observation of the molecule directly from the density data. This method is built on the use of a U-net transformer to identify C-alpha atoms from the density map and a HMM to relate them to protein sequences of multiple chains in the map. The method was tested using a large data set of 128 cryo-EM maps and another set of 500 recently-deposited density maps of 1.9-4.0 angstrom resolution. Results show that it is able to build more accurate models with more coverage of the structures than Phenix. The results presented in this paper are significant, since they show a milestone in automatic prediction of atomic structures directly from a density map.

The main contributions:

- The method is an automatic method for de novo modeling of cryo-EM density maps, unlike most modeling tools that rely on input of prior knowledge. Even though it is challenging to solve structures automatically from a cryo-EM density map that often contain multiple chains, this paper shows the potential of an automatic method.
- The use of HMM to relate C-alpha atom positions to the amino acids on multiple sequences is novel. One of the existing methods for this problem is the use of travel sales man algorithm with predetermined heuristics.
- Using a large data set, results show that Cryo2Struct has significant improvement in both accuracy and completeness of the models than for Phenix's map_to_model method to produce initial models. Since Phenix is a widely used tool, having a more accurate initial model, such as the one produced through Cryo2Struct, can potentially improve downstream refinement steps.

I have the following questions/comments.

1. The C-alpha atoms appear to be detected quite accurately, but the TM scores are low, presumably due to the alignment step. Does the HMM method consider the symmetry among multiple chains in the density map/structure? How is it different from how Phenix handles the symmetry?

Response:

Thank you for the insightful comments and suggestions. In Cryo2Struct's protein sequence alignment process for a multi-chain protein complex, the sequence of each chain is aligned with the HMM one by one. Once a chain is aligned, the states in the hidden path aligned with it are removed from the HMM before another chain is aligned. Therefore, Cryo2Struct does not consider/enforce symmetry among multiple chains in the density map, which is a limitation. We think the idea of incorporating symmetry information into the alignment process is very interesting and worth exploring. So, in the Discussion Section, we add a sentence to discuss a

future direction to consider symmetry in the model building process as follows “*The third is to leverage the symmetry of multiple chain in protein complexes to more accurately predict Ca atoms and amino acid types and align protein sequences with the HMMs*”.

In contrast, Phenix.map_to_model takes parameters such as "symmetry," "find_symmetry," and "asymmetric_map" to build structures. Consequently, Phenix can use a combination of manual input and automated detection to recognize symmetrical arrangements of density peaks or features to handle symmetry among multiple protein chains in the cryo-EM density map.

2. The introduction mentioned a few more recent deep learning methods for this problem, but the comparison was only conducted for Phenix. Can authors clarify why it is not compared with those?

Response:

Thank you for raising this issue. Indeed, there are three other related deep learning tools to build protein structures from cryo-EM density maps, which are DeepMainmast, ModelAngelo, and DeepTracer. Below are the reasons that they are not included into the comparison.

DeepMainmast uses both AlphaFold predicted structure and density maps as input for prediction and therefore is not a de novo method of building protein structures from only density map information. So, it is not included into the comparison with Cryo2Struct that uses only density maps as input to build protein structural models.

ModelAngelo uses a pretrained language model ESM-1b to generate the embeddings of a target protein as an extra input on top of density maps. The embeddings contain extra structural information about the protein. Therefore, it is not included into the comparison with Cryo2Struct.

DeepTracer primarily allows users to model structures using their web-based tool, which only permit users to make a few predictions per day. During the development and analysis of Cryo2Struct, we could not perform a large-scale modeling of density maps using DeepTracer's web server. It also provides an option to download its program under a license to be approved by the authors. But we did not receive a response from the authors to let us download its program after we applied for the license. Moreover, because the training dataset of the DeepTracer is not publicly available, we cannot remove test proteins that may be included into the training dataset of DeepTracer to fairly compare it with Cryo2Struct. For these reasons, we were not able to include it into the comparison.

Reviewer #3

In this work, the authors develop a tool for matching EM density maps to a sequence + alpha-carbon trace of the protein(s). The approach uses a transformer to identify voxels associated with the alpha carbons, and then an HMM to match sequences to the trace. The approach is quite effective and outperforms ab initio methods such as the standard Phenix platform. The tool is well described and the code made accessible both on github and codeocean for those who wish to use and/or extend the current tool.

The HMM approach determines the probability distribution of amino acids at each given Calpha based on the distribution of neighbors. This information could be converted into a confidence score - similar to the PLDDT scores used in other tools which is based on the match of a prediction to a training set of environments. Such a calculation would allow the determination of local confidence in prediction, in addition to the protein-wide metrics reported.

Response:

Thank you for your great suggestion. Following your recommendation, we have added a new subsection titled "*Confidence Scores by Cryo2Struct*" into the Results Section in the revised manuscript. We define the residue-wise confidence scores for C α atoms and amino acid types predicted by Cryo2Struct. As you pointed out, like the pLDDT scores that AlphaFold assigns to predicted structures, the confidence scores reflect the degree of confidence Cryo2Struct has in predicted C α atoms and their amino acid types, with higher scores indicating more reliable predictions, while lower scores suggest more uncertainty that warrants scrutiny of the predictions. The per-residue confidence scores for a structural model can be averaged to measure the overall confidence in the entire structural model.

We also added four new extended figures (Fig. Extended Data 7, 8, 9, and 10) to visualize and analyze the C α atom confidence scores and amino acid type confidence scores predicted for four proteins. The analysis reveals that the confidence scores of C α atom predictions are much higher than the confidence scores of amino acid type predictions. The more confident C α predictions facilitate aligning protein sequences with the HMM to build more accurate structures, but the less confident amino acid type predictions can mislead the alignment process, resulting in incorrect connections of some C α atoms.

Moreover, we have updated the source code of Cryo2Struct at GitHub to generate residue-wise confidence scores for predictions. The latest version of Cryo2Struct now not only produces atomic structures from input cryo-EM density maps but also generates an output file containing residue-wise confidence scores for both carbon-alpha atoms and their respective amino acid types automatically. The average confidence scores are also included in the output file as the protein-wide confidence score of entire modeled structures. Additionally, the code generates a scatter plot visualization for the interpretation of these confidence scores. An example of the output confidence score file is available at the GitHub repository.

We believe that these updates significantly improve the utility and interpretability of Cryo2Struct, aligning it more closely with the needs of the research community. Thank you once

again for your valuable suggestion.

Reviewers' Comments:

Reviewer #1:

Remarks to the Author:

The authors have properly addressed my concerns. There is only one minor point.

Page 9: To avoid confusion, it is better to change the section title "Evaluating Cryo2Struct on highly dissimilar structures" to "Evaluating Cryo2Struct on highly sequence-dissimilar proteins".

Reviewer #2:

Remarks to the Author:

My questions and comments were addressed by authors. I think the paper would be enhanced if it discusses beyond the initial model. The proposed method shows better accuracy than phenix.mao_to_model, yet final models built using phenix have been much more accurate than its initio models. Authors should discuss additional steps Phenix uses to refine initial models, and if they potentially benefit initial models produced from Cryo2Struct.

Reviewer #3:

Remarks to the Author:

Amino acid type and Calpha confidences appear to be uncorrelated (Extended data Figs 7-10. I would have expected the aa-type emission probability distribution to sharpen in regions of structural confidence. There also is some normalization issue as each of the 20 amino acids has a different dynamic range of confidence values (highest often attributed to Pro, Gly, Ala). Figures showing per-residue confidence values (calpha and/or aa-type) mapped onto structures using a color spectrum would be helpful to assess where in the model confidences are high/low.

While I thank the researchers for investigating per-residue aa-type confidence values, it is not clear that these scores provide useful information to assess model quality as currently presented in the manuscript.

Response to Reviewer Comments and Suggestions

We would like to express our sincere gratitude to the reviewers for their helpful suggestions and comments on our manuscript. The text changes in the revised manuscript are marked in red. Below are the point-by-point responses to the concerns raised by reviewers.

Reviewer #1

The authors have properly addressed my concerns. There is only one minor point.

Page 9: To avoid confusion, it is better to change the section title “Evaluating Cryo2Struct on highly dissimilar structures” to “Evaluating Cryo2Struct on highly sequence-dissimilar proteins”.

Response:

Thank you for the great suggestion. We have changed the section title to “Evaluating Cryo2Struct on highly sequence-dissimilar proteins”.

Reviewer #2

(1) My questions and comments were addressed by authors. I think the paper would be enhanced if it discusses beyond the initial model. The proposed method shows better accuracy than phenix.map_to_model, yet final models built using phenix have been much more accurate than its initial models. Authors should discuss additional steps Phenix uses to refine initial models, and if they potentially benefit initial models produced from Cryo2Struct.

Response:

Thank you for your great suggestions. We have added a new section called “*Refinement of modeled structures*” into the revised manuscript to discuss how to use the Phenix’s refinement function to improve the initial models built by Cryo2Struct. The Phenix’s program of building protein structural models from cryo-EM density map - Phenix.map_to_model - has included a built-in refinement step. Therefore, the structural models built by Phenix.map_to_model in this work have gone through the refinement process. As you suggested, we applied the same Phenix’s refinement function - Phenix.real_space_refinement used by Phenix.map_to_model - to refine the initial structural models built by Cryo2Struct. Interestingly, the quality of the Cryo2Struct modeled structures is notably improved by the refinement process on both the new test dataset and the standard test dataset. The detailed new results are discussed in the new section “*Refinement of modeled structures*”.

(2) I would need more time to review the details of the code. After a quick browsing of its code ocean site, I saw a ReadMe document regarding downloading and setup.

Response:

We have provide a detailed stepwise procedure for modeling atomic structure from cryoEM density maps using Cryo2Struct on Code Ocean and on GitHub repository of Cryo2Struct, available here: <https://github.com/jianlin-cheng/Cryo2Struct.git>
The GitHub also includes the training code that was used to train Cryo2Struct.

Reviewer #3

Amino acid type and Calpha confidences appear to be uncorrelated (Extended data Figs 7-10. I would have expected the aa-type emission probability distribution to sharpen in regions of structural confidence. There also is some normalization issue as each of the 20 amino acids has a different dynamic range of confidence values (highest often attributed to Pro, Gly, Ala). Figures showing per-residue confidence values (calpha and/or aa-type) mapped onto structures using a color spectrum would be helpful to assess where in the model confidences are high/low.

While I thank the researchers for investigating per-residue aa-type confidence values, it is not clear that these scores provide useful information to assess model quality as currently presented in the manuscript.

Response:

Thank you for the very insightful comments and great suggestions. Inspired by your advice, in the revised manuscript, we have completely revised the writing and methods in the Section “Confidence scores by Cryo2Struct” to address all the issues that you have mentioned.

In the previous version, we directly used the probabilities of Ca atoms predicted by deep learning as the confidence scores of Ca predictions and the emission probabilities of amino acid types from the HMM as the confidence scores of amino acid type predictions in the structural models built by Cryo2Struct. This approach indeed has several problems. *First*, the confidence scores of Ca atoms and the confidence scores of amino acid types are uncorrelated because of at least three issues: (a) the confidence score of amino acid types generated by HMM for a Ca atom is not necessarily the probability of the most likely amino acid type emitted from the hidden state representing the Ca atom in the HMM due to the use of the global Viterbi alignment algorithm for aligning the protein sequence with the Ca hidden states in the HMM, (b) the emission probability for each amino acid type in a Ca hidden state is based on the geometric mean of the probability of the amino acid type predicted by the deep learning and the background probability of the amino acid type rather than the former only, and (c) the emission probabilities for different amino acid types are in different ranges. *Second*, the probability of a Ca atom or the emission probability of an amino acid type is only related

to, but not the actual estimate of the chance (probability) of correctness, desired by a confidence score. That is, a predicted probability of 0.5 for a Ca atom does not mean the chance that the Ca atom prediction is correct is 50%. *Third*, as you pointed out, the range of the probabilities for different amino acid types is different and therefore they are not comparable. However, because the confidence scores for amino acid types should be in the same range and directly comparable, the probabilities for different amino acid types should be normalized before they can be used as confidence scores.

To address all the issues above, in this revised manuscript, we apply a new approach to estimating the confidence scores of Ca atom predictions or amino acid type predictions from their predicted probabilities. That is, we use one logistic regression to predict if a Ca atom prediction is correct from its Ca probability predicted by the deep learning method. The output of the logistic regression is the estimate of the probability that a Ca atom prediction is correct, which can serve as the confidence score of Ca atom predictions. Similarly, we use another logistic regression to predict if an amino acid type prediction is correct from both its predicted probability and the one-hot encoding of the amino acid type, which can also serve as the confidence score of amino acid type predictions. Because the one-hot encoding of the amino acid type is used as one input for the logistic regression, the logistic regression automatically generates the confidence scores for different amino acid types that are in the same range and directly comparable, solving the normalization issue.

We trained the logistic regression approach for Ca atom and amino acid type predictions respectively on a portion of the test proteins in the new test dataset and tested it on the remaining portion of the test proteins in the new test dataset that has less than 25% sequence identity with the training proteins used to train the deep learning method. The test results show that the confidence scores estimated by the logistic regression solve all the problems in the previous version.

First, there is a strong correlation between the confidence scores of Ca atom predictions and the confidence scores of amino acid type predictions. That is, the correlation coefficient between them is 0.96, with an extremely low p-value of $1.14E-92$. Second, the new confidence score can be intuitively interpreted as the chance (probability) of correctness according to the definition of the logistic regression. Third, the same confidence score value for different amino acid types means the same thing – the chance of being correct and is comparable, which solves the normalization problem.

Moreover, we investigate if the new per-residue confidence scores for Ca atoms or amino acid types are a good indicator of the true quality of the structural models built by Cryo2Struct. We use the average of the per-residue confidence scores for Ca atoms or amino acid types in a structural model as the confidence score of Ca atoms or amino acid types for the whole structural model. We calculate the correlation between the confidence scores of the structural models and their true quality scores in terms of either the Ca atom match score or the sequence match score. Indeed, there is a strong correlation between them. The Pearson's correlation coefficient between the Ca match score and the overall Ca confidence score for the 167 test structural models is 0.6, with

a corresponding p-value of 3.06E-17. The correlation between the sequence match score and the overall amino acid type confidence score for the 167 test structural models is 0.7. The results demonstrate that the confidence scores are useful for assessing the model quality.

Finally, we replace the old Extended data Figures 7-10 that plot the old confidence scores of Ca atom predictions and amino acid type predictions with a new Figure Extended Data 7 show an example plotting the new confidence scores of C α atoms against the new confidence scores of amino acid types for a test protein, which indicates a robust positive connection between the two kinds of confidence scores.

We also add a new Figure Extended Data 8 showing two test examples of visualizing amino acid type confidence scores on top of the Cryo2Struct modeled structures using a color spectrum and a new Figure Extended Data 9 using a detailed test example to demonstrate how different confidence scores can help users to identify high- or local-quality regions in the structural model in comparison with the true structure. The examples demonstrate that the confidence score can indicate the high/low local quality of each residue in a Cryo2Struct modeled structure, as pLDDT score is used to indicate the quality of each residue in an AlphaFold predicted structure.

All the new methods and results above have been added into the Section "Confidence scores by Cryo2Struct".

Reviewers' Comments:

Reviewer #2:

Remarks to the Author:

Authors have addressed my comments.

Reviewer #3:

Remarks to the Author:

I thank the authors for considering the construction of a normalized local confidence metric that can be mapped onto the structure. I have no further concerns with this study.